# A Rapid Review Exploring the Role of Yoga in Healing Psychological Trauma

**DOI:** 10.3390/ijerph192316180

**Published:** 2022-12-03

**Authors:** Arabella English, Elizabeth McKibben, Divya Sivaramakrishnan, Niamh Hart, Justin Richards, Paul Kelly

**Affiliations:** 1Edinburgh Medical School, The University of Edinburgh, Edinburgh EH16 4TJ, UK; 2Faculty of Health, Victoria University of Wellington, Wellington 6140, New Zealand; 3Physical Activity for Health Research Centre (PAHRC), The University of Edinburgh, Edinburgh EH8 8AQ, UK

**Keywords:** yoga, psychological trauma, impact, rapid review, qualitative literature

## Abstract

The evidence regarding the benefits of yoga for treating psychological trauma is well-established; however, there is a paucity of qualitative reviews exploring this topic. The purpose of this rapid review is to gain a deeper understanding of the impact that yoga can have on people with a history of psychological trauma and to reveal barriers and facilitators to the uptake of yoga in this cohort, from a qualitative perspective. The Ovid(EMBASE), Ovid(MEDLINE), PsycINFO, PubMed, and SPORTDiscus databases were searched using key terms. The systematic search generated 148 records, and 11 peer-reviewed articles met the inclusion criteria. The following main impacts of yoga on participants were identified: feeling an increased sense of self-compassion; feeling more centred; developing their coping skills; having a better mind–body relationship; and improving their relationships with others. The main barriers were also identified: concerns initiating yoga; time and motivational issues; and the costs and location of classes. The main facilitator was the feeling of safety generated in the trauma-informed yoga classes. This review suggests that yoga offers great potential in the field of trauma recovery. Despite this, more high-quality research with rigorous methodologies is called for to allow this field to advance.

## 1. Introduction

Psychological trauma is a complex emotional response to a deeply distressing or disturbing experience. It is estimated that most individuals are exposed to a traumatic event at some point in their lifespan. Indeed, an analysis of WHO World Mental Health surveys across 24 countries found that 70.4% of respondents had experienced some sort of traumatic event during their lifetime, with an average of three specific lifetime traumas per person [1]. Trauma exposure is a known risk factor for many psychiatric illnesses. For example, trauma is associated with PTSD and other mood, anxiety, and substance abuse disorders [2]. The burden this can have on already overstretched and underfunded mental health services is significant, and the need for effective and pragmatic treatment options is extremely important.

Psychotherapy is often considered the ‘gold standard’ of psychological trauma treatment [3]. However, patients who are resistant to talk therapies and cognitive-based interventions often disengage from treatment [4]. Treatments that take a more holistic approach may help to retain engagement and improve outcomes. There is a large body of research that associates physical activity with positive mental health outcomes, in both general and trauma-exposed populations [5,6].

Yoga is a 3000-year-old mind–body–spirit practice which combines physical movement with mindful focus on internal awareness of one’s breath, energy, and self. The word “yoga” emerges from the sanskrit root word of “yuj”, meaning “to yoke,” and is often translated in the contemporary context as “union.” While this spiritual union shapes the foundations of yoga, contemporary yoga is often conceptualised as a physical wellbeing exercise [7]. However, the other components of breathwork and meditation are just as important in generating a healing process for the practicing individual. These components work synergistically to alleviate stress, cultivate mindfulness, and boost physical and mental wellbeing. The impact that this practice can have on individuals with a history of psychological trauma is of increasing interest and importance [8]. A systematic review and meta-analysis by Taylor et al. [8] found that yoga produced a moderate effect (g = 0.46, *p* < 0.001) on reducing trauma-related symptoms in people who have been exposed to psychological trauma, alongside clinical approaches. Thus, yoga may be able to address the holistic needs of trauma survivors in a way that psychotherapy cannot.

Despite there being a well-established evidence base that acknowledges the benefits of yoga for treating psychological trauma [8,9,10,11], there is a paucity of research synthesising the qualitative reviews that explore this topic. Qualitative research can help to produce a more in-depth understanding of human behaviour, attitudes, intentions, and motivations, and can add meaningful context and nuance to quantitative literature. The aims of this review are therefore to reveal and deepen the understanding of the impact that practising yoga can have on the mental wellbeing of people with psychological trauma from a qualitative perspective. The review also aims to explore barriers and facilitators to the uptake and delivery of yoga as an intervention for people who have a history of psychological trauma, and to identify key evidence gaps and research priorities in this area. From this, health professionals involved in treating trauma-exposed patients will have a more in-depth understanding of the holistic treatment options available and will potentially be able to incorporate them into their services. Furthermore, yoga instructors providing trauma-informed classes will be better able to optimise their delivery of services to maximise the positive effects for participants. 

## 2. Materials and Methods

A rapid review (RR) utilises simplified systematic review methods to provide a snapshot of the literature in a more timely manner. A simplified methodological process is utilised to allow for a formal quality assessment of the literature to be made [12]. Various RR methodologies have been described in the literature, and guidance was therefore sought from the review by Haby et al. [13]. Our review was guided by the five-stage framework proposed by Arksey et al., as outlined below [14].

Stage 1—Identifying the research questions

The following main research questions for the project were identified in the preliminary discussions had within the research team, based on current gaps in the literature:What is the nature of the evidence base in exploring the role of yoga in healing psychological trauma?What is the impact of practising yoga for people with a history of psychological trauma?What are the main barriers and facilitators to the uptake and delivery of yoga for people who have had a history of psychological trauma?

Stage 2—Identifying relevant studies

The search strategy involved searching the following databases: Ovid(EMBASE), Ovid(MEDLINE), PsycINFO, PubMed, and SPORTDiscus, with the search terms depicted in Table 1. Searches were run in January 2022.

The databases outlined above were searched for titles and abstracts using the key words stated in Table 1, combined with the Boolean operators “and” (within categories) and “or” (between categories).

Stage 3—Study selection

All identified studies were uploaded to Covidence (https://www.covidence.org), where duplicates were automatically removed. All titles and abstracts were then screened and excluded if appropriate. The remaining studies were examined as full texts by two authors (A.E. and E.M.), with reference to the eligibility criteria (Table 2), and the final studies to be included in the review were identified. 

Stage 4—Charting the data

Data were extracted by the lead author (A.E.) with a data extraction form (Appendix A). The form was piloted using the first 10% of studies to ensure that the approach to data extraction was consistent with the research questions. When full papers could not be found, efforts to obtain a hard or electronic copy from the university library were made.

Stage 5—Collating, summarising, and reporting the results

The results were presented in two ways:A descriptive analysis to provide a comprehensive overview of the prevailing qualitative research regarding yoga and psychological trauma.A narrative summary of the main themes explored in the studies.

## 3. Results

Research question 1—What is the nature of the evidence base in exploring the role of yoga in healing psychological trauma?

In total, 148 records were identified through the systematic search. Following the removal of duplicates, title and abstract screening, and full-text screening, a total of 11 studies were included in the review that met the eligibility criteria (Figure 1).

Most of the studies took place in the USA (*n* = 9), with one study being conducted in South Africa, and one in Canada. The studies were published between 2015 and 2022, and all were peer-reviewed.

Of the 11 studies included in the review, 8 utilised a qualitative research design [15,16,17,18,19,20,21,22]. Six of these qualitative studies employed the use of semi-structured interviews, two studies used structured interviews, and one used an online survey. The other three studies had a mixed-methods design [23,24,25], one of which involved semi-structured interviews, and the other involved in-person questionnaires. Seven of the studies were observational in nature, and four provided qualitative analysis of larger randomised controlled trials by interviewing or surveying participants who were in the yoga intervention group as opposed to the control group. Five of the studies recruited female-only participants and the other six had a mixture of male and female participants.

Eight studies explored the impact that trauma-sensitive yoga (TSY) has on healing psychological trauma [15,17,18,19,20,21,23,25]. Two studies used non-TSY yoga interventions, with one looking specifically at Ashtanga yoga [24] and the other at Kundalini yoga [16], and one study did not specify the style of yoga used [22]. TSY is a yoga intervention that is aimed towards individuals with a history of trauma exposure. A safe environment and gentle teaching approach enable participants to address specific trauma-related symptoms that they may suffer from [26]. Ashtanga and Kundalini yoga are other popular styles of yoga taught in the general yoga community. Other key information regarding the included studies is summarised in Appendix A.

Summary: There is a growing body of qualitative evidence centred around the healing effects of yoga for individuals with psychological trauma, and around the facilitators and barriers associated with the uptake of yoga classes. Most of this research is based in the USA and has been conducted over the past seven years. TSY is the most popular intervention, but other styles of yoga have also been explored.

Research question 2—What is the impact of practising yoga for people with a history of psychological trauma?

Self-compassion

One finding seen across many of the studies was that participants reported increased self-compassion since starting their yoga practice [15,16,18,22,23]. Participants in studies by West et al. [15] and Gulden et al. [22] found that the emphasis on moving away from self-judgement helped transform their perceptions of themselves off the mat. In a study by Crews et al. [18], participants noted that partaking in the yoga class was a way of rewarding themselves with kindness and self-care.

Centredness

Many participants reported feeling more centred since starting yoga intervention [15,19,20,25]. In a study by Schmid et al. [19], participants noted that the act of setting an intention at the start of each yoga class allowed them to have an improved sense of focus and inner calmness. This helped them to alter their mindset to a calm and relaxed state off the mat. Participants in the study by West et al. [15] noted having a quieter mind, less rumination, an ability to see different perspectives, and a generally more positive outlook since starting yoga.

Coping skills

Participants across several studies found that the yoga practice allowed them to develop more effective coping tools and mechanisms [16,17,23,24]. Studies by Braun et al. [23] and Rhodes [17] found that participants were inclined to use the yoga postures and breathing exercises as coping mechanisms when their stress and trauma triggers arose in their day-to-day lives.

Mind–body relationship

A few of the papers in the review touched on the effect that the physical aspect of the yoga practice had on participants [20,22,23,24,25]. In a study by LaChiusa [24], participants commented on how certain parts of the body hold on to memories and strong emotions, such as fear, anxiety, and anger. They found that certain hip- or back-opening postures tended to bring up intense feelings. Over time, however, this intensity faded, which resulted in participants having a greater feeling of control over their emotions.

Interpersonal relationships

Several studies commented on the positive impact that the yoga intervention had on participants’ interpersonal relationships [16,17,18,20,21,23,25]. In a study by Cushing et al. [20], participants specified how PTSD symptoms can be isolating and have prevented them from trying new activities. Knowing that everyone else partaking in the yoga intervention had also been through similar experiences helped participants to feel more comfortable, as it provided a safe space for them to explore their yoga practice. Crews et al. [18] also reached similar findings. Participants found that practising yoga as a group of women who had a shared experience of being survivors of sexual violence helped them to build better connections with each other. The yoga allowed women to bond on a ‘mind-body-spirit’ level, as opposed to the bonding achieved in talk-therapy groups, and it enabled the women to feel the support and acceptance from being part of a healthy community.

Summary: The yoga interventions impacted participants in various meaningful ways. Across the studies included in this review, the main effects that a yoga practice had on participants with psychological trauma were increased self-compassion, feeling more centred, improved coping skills, a better mind–body relationship, and enhanced relationships with others.

Research question 3—What are the main barriers and facilitators to the uptake and delivery of yoga for people who have had a history of psychological trauma?

Barriers

Concerns from participants in initiating a yoga practice were addressed in two of the studies [20,23]. Both studies involved US veterans, which shone a light on the potential stigma surrounding holistic practices in this population. In a study by Cushing et al. [20], some of the male veterans described yoga as socially unacceptable for them and physically unchallenging. One participant noted that yoga is about being open and vulnerable, which can be hard for veterans who have been trained to stay extremely vigilant and on guard.

Lack of time and motivation was brought up in three of the papers [17,23,25]. In the study by Justice et al. [25], where the intervention involved group classes as well as a shorter home practice, all participants noted the feeling that they did not have enough time to practise at home. There was also a shared feeling of having decreased motivation to maintain a home practice, without the group setting and structured class.

Two of the studies commented on the cost and location of yoga classes as barriers to participation [17,25]. In the study by Rhodes [17], participants found that the cost of classes would deter them from attending yoga classes in the community. Moreover, it was made apparent that community TSY classes may be too far from home and therefore not accessible.

Summary: The following barriers to the uptake and delivery of yoga were identified for people with a history of trauma: concerns initiating yoga; time and motivational issues; and the cost and location of classes.

Facilitators

The importance of feeling safe in the yoga class was addressed in two of the papers [17,23]. In the study by Braun et al. [23], participants noted a ‘sense of sisterhood’ and ‘level of safety’ since they were practising yoga with other survivors of military sexual trauma. They also commented that they appreciated the lack of hands-on adjustments in the TSY intervention. Participants in the study by Rhodes [17] also felt that the TSY classes provided a gentle choice-orientated approach compared to community yoga classes.

One study [25] found that all participants enjoyed the philosophical discussion that took place at the weekly in-person feasibility questionnaires and in the follow-up interviews. Participants thought it was interesting to hear about the background of yoga and to hear others’ perspectives. Some were also able to draw parallels between their religious beliefs and the yoga philosophy.

Summary: Fewer facilitators were identified in the included studies, as compared to barriers. The feeling of safety achieved in a trauma-informed yoga class was the main facilitator discussed.

## 4. Discussion

To our knowledge, this is the first review of qualitative research investigating the impact of yoga on those who have experienced trauma. Yoga is widely accepted to have a positive impact on mental health [27,28]. Yet, little is known about the experiences of yoga in the treatment of psychological trauma. This review explores the application of qualitative research to uncover the thoughts, beliefs, and experiences surrounding TSY. The following discussion elaborates on the outcomes, barriers, and facilitators of participation in TSY, and identifies pathways for theoretical development in qualitative research on yoga.

From the studies included in this review, the major outcomes that participants reported following their yoga intervention were of increased self-compassion; the feeling of being more centred; improved coping skills; a better mind–body relationship; and enhanced relationships with others. The therapeutic mechanisms underlying the positive effects that yoga can have on trauma-exposed populations are not entirely clear. All three of the main components involved in a yoga class, namely breathing exercises, physical postures, and mindfulness meditation, have been shown to affect neurobiological functioning [29]. Through the practice of mindfulness, individuals are encouraged to address potentially distressing and upsetting thoughts and emotions in a non-judgemental way, which may serve to provide an element of exposure therapy [10]. The physical aspect of yoga facilitates the observation and toleration of physical sensations [30]. Through practice, this can give individuals the power to disconnect physical feelings from emotional reactions, which can offer a turning point in healing from psychological trauma. In addition to this, attending a yoga class enables individuals to get out of the house and engage with other people, which can help consolidate a sense of self and belonging in the world [31].

The findings of this review are consistent with critiques of yoga interventions. Specifically, yoga interventions do not provide enough detail around the postures and sequences of asana to generate consistent and comparable results [32,33,34]. Of the articles identified for this review, only one provided a comprehensive outline of the postures in each class throughout the duration of the intervention [25]. TSY is designed specifically for trauma, and thus a more precise understanding of asana sequencing may help to differentiate this practice from other forms of yoga. Attending to these details can further enhance research by identifying the relationships between asana, affectivity, and safe spaces.

Across this rapid review, researchers were attentive to providing safe spaces for people to practise yoga. All studies affirmed that the teachers were certified yoga teachers, some with TSY-specific training. While accreditation processes in the yoga industry attempt to create minimum standards for teaching, these teachers are individuals that bring their own unique teaching styles and personalities to their work. Having multiple teachers lead yoga classes in a single intervention [21] introduces another layer of contextual subjectivity to TSY interventions. Indeed, relationships and perceptions of teachers can shape yoga students’ experiences of depression and anxiety [35]. Considering the personhood of the teacher in the context of the intervention may be significant in order to understand a participant’s subjective experiences. For example, in the case of Cushing’s work [20], knowing that their yoga teacher had similar experiences of war provided comfort and increased feelings of safety for the participants within the yoga class. Attending to the subjectivities of both the participants and the yoga teachers may help to enhance the contextual understandings of the impact of TSY yoga.

This review also outlined potential barriers and facilitators to the uptake of yoga classes in the trauma-exposed population. These barriers were concerns about initiating yoga, time and motivation, and the costs and location of classes, whilst facilitators included a sense of safety in the classes and provided space for philosophical discussions. Yoga instructors providing classes to trauma-exposed populations can use this insight to reflect on and improve the service that they offer. For example, offering male- or female-only classes or free trials could potentially help to maximise their service provision for individuals who are in need. This agrees with findings from our previous work (EM) that show that stigma surrounding yoga is a significant barrier to yoga uptake in both the general population and in yoga-based research [36]. This should be prioritised in future research in the field of yoga so that steps can be made to overcome this important obstacle.

In addition to future developments in the practice of delivering yoga interventions, this review also sheds light on the need for further theoretical development in qualitative research on yoga. Qualitative research is useful for gathering experiential and contextual data [37]. For this reason, the social and cultural contexts of research are important for interpreting subjective meanings. While the articles in this review attend to subjective experiences of trauma, less attention is given to the spatial, relational, and cultural contexts that shape the TSY interventions. For example, the studies included in this review involved a mixture of female-only and mixed-gender studies. This did not consider the gender- or sex-based differences of yoga, such as anatomical variations and societal attitudes attached to practising yoga. In addition, there was disparity in whether participants were receiving concurrent psychological treatment, either as part of the study, or of their own accord.

In addition to the importance of relational context, spatial context has a significant impact on health [38]. Theories of emplacement articulate how bodies are connected to the socio-cultural meaning of the spaces where they engage in physical activity [39]. Within this review, only two articles attended to the type of space where yoga was taking place. Crews et al. [18] address how their yoga classes take place in a local centre for trauma survivors, whilst the survey in the study by Gulden et al. [22] asked participants where they practise (for example, a studio or community centre). Despite introducing these spaces, little discussion dives into the significance of these localities. The socio-cultural meanings associated with any one space may also influence the subjective experiences of the participants. Future research could attend more to the subjective experiences of yoga in space, both on a micro and macro level.

Macro-level socio-cultural contexts likewise play a part in the subjective meanings made through yoga. Within this review, two articles broadly made reference to the geopolitical spaces of the study, with only one discussing the impact of this context on trauma [16]. In this article, researchers articulate the high percentage of trauma experiences in South Africa and point to the utility of yoga in healing from traumatic experiences. While participants in this research lauded yoga for being a safe space for healing, yoga practices may not always be a safe haven from trauma. For example, yoga’s history of sexual violation is often overlooked because of its curative impacts. Despite the allegations of sexual abuse against prominent teachers, many yoga lineages continue to deliver their curriculums without interruption [40]. Understanding the potential of TSY for healing must also be understood in the socio-cultural contexts of yoga’s hierarchical and, at times, violent history. For the many who have experienced trauma through yoga, TSY may not be the panacea it appears to be.

Strengths and Limitations

The conclusions of this review are, to an extent, limited by the number of studies that it was able to analyse. Many of these were mixed-method studies and operated under fairly positivist frameworks that do not consider intervention contexts holistically. In addition, the studies included in this review involved a mixture of female-only and mixed-gender studies. This did not consider the gender- or sex-based differences of yoga, such as anatomical variations and societal attitudes attached to its practice. In addition, there was disparity in whether participants were receiving concurrent psychological treatment, either as part of the study, or of their own accord. It is also important to consider the fact that due to time constraints and the nature of a rapid review, less stringent methods were used, as compared to systematic reviews. For example, non-English language studies were excluded, which limits the generalisability of results to non-English-speaking counties.

Despite this, the review does still provide novel insight into this area of research, as it is the first of its type to explore the qualitative literature across a range of psychological traumas. Implementing the rapid review methodology helped generate applicable insight. However, given these preliminary findings and the limitations outlined above, future research should implement more focussed qualitative assessments in trauma-exposed populations. The comparison of a yoga-only and yoga-plus-psychotherapy intervention could provide a greater depth of understanding into the exact role that yoga could play in treatment options. Furthermore, once more evidence is available, more detailed reviews can be undertaken. Future research should try to generate replicable guidelines that can be translated into clinical practice, for example, in teaching styles, sequences, and class structures. In addition, while yoga can be considered accessible, it is important to remember that not all people who have experienced psychological trauma have the physiological or mental capacity to take part. Therefore, the findings here may not be applicable to all individuals.

## 5. Conclusions

This study sought to identify and understand the qualitative evidence of yoga as a treatment for psychological trauma. The identified studies indicate that yoga may have an important role to play in the field of trauma recovery, equipping individuals with the tools to claim control over their symptoms. Despite this, more high-quality research with rigorous methodology is called for to allow this field to advance and to have greater implications for clinical practice.

## Figures and Tables

**Figure 1 ijerph-19-16180-f001:**
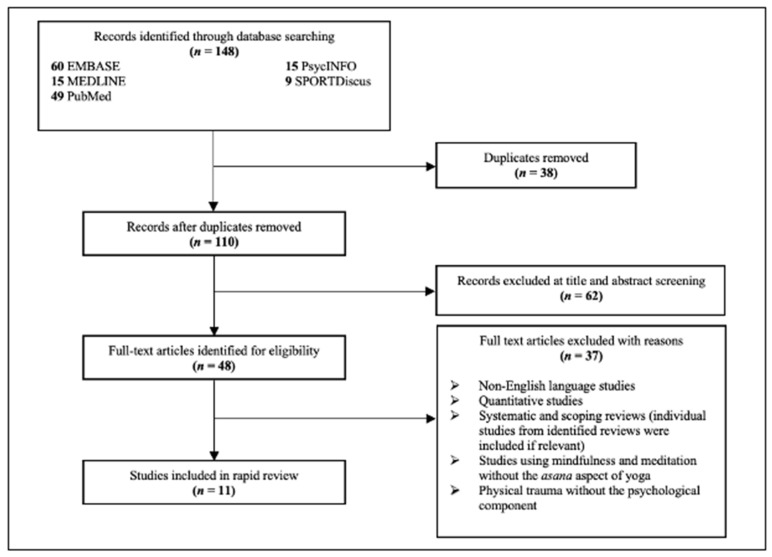
Results of the systematic electronic search.

**Table 1 ijerph-19-16180-t001:** Key search terms.

Yoga Keywords	Study Design Keywords	Psychological Trauma Keywords
**Key terms:** Yog* (yoga, yogi, yogic) Asana	**Key terms:** Qualitative Focus group Interview*	**Key terms:** Psychological trauma Post-traumatic stress disorder PTSD Violence Abuse Neglect Multiple trauma Sexual trauma

**Table 2 ijerph-19-16180-t002:** Eligibility criteria.

Inclusion Criteria	Exclusion Criteria
All geographical locations and settings Any age groups and sex Human studies Yoga as an intervention for those suffering from psychological trauma Yoga as an adjunctive treatment for those receiving treatment for psychological trauma All types of yoga Qualitative studies	Non-English-language studies Quantitative studies Systematic and scoping reviews (individual studies from identified reviews were included if relevant) Studies using mindfulness and meditation without the *asana* aspect of yoga Physical trauma without the psychological component

## Data Availability

Not applicable.

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
