# Peer review of "A Rapid Review Exploring the Role of Yoga in Healing Psychological Trauma"

_ijerph, 2022, doi:10.3390/ijerph192316180_

Round 1

Reviewer 1 Report

The paper is fine example of a rapid review study; what it promises it delivers. Considering the argument about stigma surrounding yoga (lines 289-292), a paper which does not include one of the authors would be better suited (reference 36).

Author Response

  1. The paper is fine example of a rapid review study; what it promises it delivers. 

Many thanks for this comment.

  1. Considering the argument about stigma surrounding yoga (lines 289-292), a paper which does not include one of the authors would be better suited (reference 36).

We have reflected that this is a fair comment. To further investigate we did an incognito boolean search for “yoga” AND “mental illness” AND “stigma” to try and identify new studies without bias from our previous searches. However, this particular article was the first to come up and the other papers are not quite on this issue would not be useful in supporting (or refuting) the point we are trying to make.

But we agree that the wording was not quite right and have updated in manuscript to: "This agrees with findings from our previous work (EM) that stigma surrounding yoga is a significant barrier to yoga uptake in both the general population and in yoga-based research [36]."

We feel this is a more balanced statement and now more transparent for the reader.

Reviewer 2 Report

Dear Authors:

I think your manuscript it is important and interesting in search of a holistic medicine, more over nowadays on XXI century and US, when and where, material conditions are on. Allopathic, physiological and psychological, medicine must manage to integrate the best possibilities to achieve better quality and scalability in the health of patients. In this sense, your proposal for publication is interesting and worth publishing. But I have some details to share with you for a better and greater positive impact in favor of this integration.

One big issue must be the incapacity by physiological or mental conditions to do the Yoga.

Another methodological recommendation is that Covidence use different types of reviews, for qualitative research the main core of information it is about subjectivity, social and cultural symbolism therefor the best review is the “Literature reviews or narrative reviews” (LRNR)...in fact, for methodological precision it could be nice to explain which kind of review you used and if it was not the LRNR, it could be great if you explain us which one you use and why? And how? this option of review influences on the results.

In the introduction, more details are missing about what and how the research was done, but above all about which is the aim of the research, I read about on lines 16-21, I suggest putting it on the introduction: “The main themes identi-16 fied were that yoga resulted in participants feeling an increased sense of self-compassion; feeling 17 more centred; developing their coping skills; having a better mind-body relationship; and improv-18 ing their relationships with others. This review suggests that yoga offers great 21 potential in the field of trauma recovery.” But also I suggest to put above the objectives of the research, meanwhile you put it down on lines 55-65.

Unless you recover valuable information about facilities and barriers for using Yoga as a potential practice for health, it is highly recommend to focus as a practice for therapeutic Psychological and physiological Trauma, I mean as a part of a holistic system or process. About this specific relation between the trauma and Yoga, I read Just a little info on lines 320-322, it could be great if you tell us more.

About redaction I recommend you write another classification on Table 1. Key search terms, specifically on third column because PTSD,  Violence, Abuse and Neglect are not Psychological Traumas. And I found just one type Error on 311.. centre instead of center.

Author Response

  1. I think your manuscript it is important and interesting in search of a holistic medicine, more over nowadays on XXI century and US, when and where, material conditions are on. Allopathic, physiological and psychological, medicine must manage to integrate the best possibilities to achieve better quality and scalability in the health of patients. In this sense, your proposal for publication is interesting and worth publishing.

Many thanks to the Reviewer for this positive feedback.

But I have some details to share with you for a better and greater positive impact in favor of this integration.

  1. One big issue must be the incapacity by physiological or mental conditions to do the Yoga.

Thank you for this point. It is important to consider the fact that certain people may have limitations to their practice due to physiological or mental conditions. However, part of the beauty of yoga is that each person’s practice can look a little bit different, and posture modifications and props can be adopted. There are many adaptations and yoga styles available, such as chair yoga, yoga for seniors and more mindfulness-based practices for those with other limitations. There has been a rise in the ‘trauma informed’ yoga that has been referenced in this review and mental health-specific yoga due to the influx of research that supports its positive impact on affected individuals. While some people may be unable to practice some styles of yoga based on their physiological or mental conditions, the yoga described in this study is specifically catered to those needs.

However, we agree the reviewer has raised an important point and have added the following to the Discussion (line 367-370).

“In addition, while yoga can be considered accessible, it is important to remember that not all people who have experienced psychological trauma have the physiological or mental capacity to take part. Therefore, the findings here may not be applicable to all individuals.”

  1. Another methodological recommendation is that Covidence use different types of reviews, for qualitative research the main core of information it is about subjectivity, social and cultural symbolism therefor the best review is the “Literature reviews or narrative reviews” (LRNR)...in fact, for methodological precision it could be nice to explain which kind of review you used and if it was not the LRNR, it could be great if you explain us which one you use and why? And how? this option of review influences on the results.

Thanks for this comment, and we agree with the reviewer. We feel we have appropriately identified this study as a rapid review in the title, abstract (line 12), keywords, and start of methods (line 73 – with reference to Grant et al., 2009 Typology of Reviews). The methods state:

“A rapid review (RR) utilises simplified systematic review methods to provide a snapshot of the literature in a more timely manner. A simplified methodological process is utilised to allow for a formal quality assessment of the literature to be made [12]. Various RR methodologies have been described in the literature, and guidance was therefore sought from the Haby et al. review [13].”

We feel this already addresses the comment, and as such we have not made any changes to the manuscript.

Grant MJ, Booth, A. A typology of reviews: an analysis of 14 review types and associated methodologies. Health Information & Libraries Journal. 2009; 26(2): 91-108. DOI: 10.1111/j.1471-1842.2009.00848.x

  1. In the introduction, more details are missing about what and how the research was done, but above all about which is the aim of the research, I read about on lines 16-21, I suggest putting it on the introduction: “The main themes identi-16 fied were that yoga resulted in participants feeling an increased sense of self-compassion; feeling 17 more centred; developing their coping skills; having a better mind-body relationship; and improv-18 ing their relationships with others. This review suggests that yoga offers great 21 potential in the field of trauma recovery.” But also I suggest to put above the objectives of the research, meanwhile you put it down on lines 55-65.

Thank you for this comment. In our opinion, lines 11-14 in the Abstract reflect the aims of the research, as outlined in lines 55-65.

“The aims of this review are therefore to reveal and deepen understanding of the impact that practising yoga can have on the mental wellbeing of people with psychological trauma from a qualitative perspective. The review also aims to explore barriers and facilitators to the uptake and delivery of yoga as an intervention for people who have a history of psychological trauma and to identify key evidence gaps and research priorities in this area.”

We hope that this explanation is helpful to the reviewer.

  1. Unless you recover valuable information about facilities and barriers for using Yoga as a potential practice for health, it is highly recommend to focus as a practice for therapeutic Psychological and physiological Trauma, I mean as a part of a holisticsystem or process. About this specific relation between the trauma and Yoga, I read Just a little info on lines 320-322, it could be great if you tell us more.

We have added some extra detail (lines 335-338) to reflect this comment.

“In this article researchers articulate the high percentage of trauma experiences in South Africa and point to the utility of yoga in healing from traumatic experiences. While participants in this research lauded yoga for being a safe space for healing, yoga practices may not always be a safe haven from trauma.”

  1. About redaction I recommend you write another classification onTable 1. Key search terms, specifically on third column because PTSD, Violence, Abuse and Neglect are not Psychological Traumas. 

We have included these search terms as long-term psychological damage can occur following these experiences in life. The studies that we have included in the paper involve exploring the impact of yoga on people with psychological trauma, many of whom have experienced violence and abuse in their past which has contributed to their current state. For example, a number of the papers focused on yoga for veterans, who have psychological trauma as a result of their experience at war. We hope that this makes sense to the reviewer.

  1. And I found just one type Error on 311.. centre instead of center.

Thank you, we have edited this (line 321 and 328).

Reviewer 3 Report

I' ve been studying to become yoga teacher for three years, and practicing yoga since I was 25. It works definitely to embrace the biological, psychological, social, and existential approaches to Health. It brings health, it is able some how to put in silence the ruminating thinking which might be consequences of what the doctors call trauma, the yoga teachers calls simply the mind. By paradox,  "There is no trauma or better everyone is born with it. But according to the schools (Patanjiali) it is the mind that causes this obsession, and the scope of the yoga is to clear the mind "which is like a monkey which jumps from one branch to another one of the trees ". The concept of ASANA is standing still in a regal position: that standing still would be able to clean that jumping of the mind. I think that in this article  the basic common philosophical roots are missing, and on the contrary, it is just put how yoga is considered (valid/  not valid), that there are (hopefully) many different approaches. )  But here is the common root why yoga is here, and created as a gift of wellbeing, of accepting whatever reality it is out there, because traumas are always there, in life and for sure in dying (thinking to the finitude of the human being). THE ASANAS and especially the meditation part is that BODY posture in which the mind concentrates on breathing comes to discover while leaving the obsessive thinking different solutions and it comes nearer to personal meaning.  This meaning is the therapeutic action. As far as I'm concerned Yoga is a gift and should be taught everywhere. Now some parts of yoga are just newly branded with the terms MIndfulness. But mindfulness is nothing less than a part of yoga. 

In synthesis, I suggest to the author to better go inside the roots of yoga before speaking about its "clinical use". 

Author Response

  1. I've been studying to become yoga teacher for three years and practicing yoga since I was 25. It works definitely to embrace the biological, psychological, social, and existential approaches to Health. It brings health, it is able somehow to put in silence the ruminating thinking which might be consequences of what the doctors call trauma, the yoga teachers calls simply the mind. By paradox, "There is no trauma or better everyone is born with it. But according to the schools (Patanjiali) it is the mind that causes this obsession, and the scope of the yoga is to clear the mind "which is like a monkey which jumps from one branch to another one of the trees". The concept of ASANA is standing still in a regal position: that standing still would be able to clean that jumping of the mind. I think that in this article the basic common philosophical roots are missing, and on the contrary, it is just put how yoga is considered (valid/not valid), that there are (hopefully) many different approaches. But here is the common root why yoga is here, and created as a gift of wellbeing, of accepting whatever reality it is out there, because traumas are always there, in life and for sure in dying (thinking to the finitude of the human being). THE ASANAS and especially the meditation part is that BODY posture in which the mind concentrates on breathing comes to discover while leaving the obsessive thinking different solutions and it comes nearer to personal meaning.  This meaning is the therapeutic action. As far as I'm concerned Yoga is a gift and should be taught everywhere. Now some parts of yoga are just newly branded with the terms Mindfulness. But mindfulness is nothing less than a part of yoga. 

Thank you for this interesting insight.

  1. In synthesis, I suggest to the author to better go inside the roots of yoga before speaking about its "clinical use". 

We agree and have added some more discussion about the roots of yoga in the introduction, lines 43-51.

“Yoga is a 3,000-year-old mind-body-spirit practice which combines physical movement with mindful focus on internal awareness of one’s breath, energy, and self. The word “yoga” emerges from the sanskrit root word “yuj” meaning “to yoke,” and if often translated in the contemporary context as “union.” While this spiritual union shapes the foundations of yoga, contemporary yoga is often conceptualized as a physical wellbeing exercise [7]. However, the other components of breathwork and meditation are just as important in generating a healing process for the practising individual. These components work synergistically to alleviate stress, cultivate mindfulness, and boost physical and mental wellbeing.”